# Artemisinin-independent inhibitory activity of *Artemisia* sp. infusions against different *Plasmodium* stages including relapse-causing hypnozoites

Kutub Ashraf[1,2,3,*] , Shahin Tajeri[1,*] , Christophe-Sébastien Arnold[4] , Nadia Amanzougaghene[1],
Jean-François Franetich[1], Amélie Vantaux[2,3] , Valérie Soulard[1], Mallaury Bordessoulles[1], Guillaume Cazals[5],
Teun Bousema[6], Geert-Jan van Gemert[6] , Roger Le Grand[7], Nathalie Dereuddre-Bosquet[7], Jean-Christophe Barale[3,8] ,
Benoit Witkowski[2,3], Georges Snounou[7], Romain Duval[9], Cyrille Y Botté[4] , Dominique Mazier[1]

Artemisinin-based combination therapies (ACT) are the frontline treatments against malaria worldwide. Recently the use of traditional infusions from *Artemisia annua* (from which artemisinin is obtained) or *Artemisia afra* (lacking artemisinin) has been controversially advocated. Such unregulated plant-based remedies are strongly discouraged as they might constitute sub-optimal therapies and promote drug resistance. Here, we conducted the first comparative study of the anti-malarial effects of both plant infusions in vitro against the asexual erythrocytic stages of *Plasmodium falciparum* and the pre-erythrocytic (i.e., liver) stages of various *Plasmodium* species. Low concentrations of either infusion accounted for significant inhibitory activities across every parasite species and stage studied. We show that these antiplasmodial effects were essentially artemisinin-independent and were additionally monitored by observations of the parasite apicoplast and mitochondrion. In particular, the infusions significantly incapacitated sporozoites, and for *Plasmodium vivax* and *P. cynomolgi*, disrupted the hypnozoites. This provides the first indication that compounds other than 8-aminoquinolines could be effective antimalarials against relapsing parasites. These observations advocate for further screening to uncover urgently needed novel antimalarial lead compounds.

# Introduction

Sustainable control and elimination of malaria mainly relies on effective treatment, as well as on the ability to eliminate the dormant forms (hypnozoites) responsible for the relapses of the widely distributed *Plasmodium vivax* (Mazier et al, 2009). Artemisinin-based combination therapies (ACT) are the frontline treatments against malaria worldwide (WHO, 2019b). The recent emergence and spread of ACT-resistant *Plasmodium falciparum* strains in South-East Asia, is of major concern, and a similar emergence in Africa will have catastrophic consequences (Imwong et al, 2017; Lu et al, 2017; Uwimana et al, 2020). Elimination of hypnozoites, the dormant hepatic forms responsible for *P. vivax* relapses, can only be achieved at present using 8-aminoquinoline drugs (primaquine or tafenoquine), but their use is restricted by their deleterious effects in persons with glucose-6-phosphate dehydrogenase deficiency (Mazier et al, 2009). Thus, novel antimalarials are urgently needed.

Artemisinin present in *Artemisia annua* (sweet wormwood) is considered to be solely responsible for the plant's potent activity against the parasite's blood stages. Recently, decoctions of *Artemisia*-dried leaves have been controversially advocated as cheaper traditional plant-based treatments for malaria. Such self-administered and unregulated treatments are strongly discouraged because of variable and potentially sub-optimal artemisinin content not least as they would promote the emergence of drug resistance (WHO, 2019a). It has been suggested that the efficacy of

[1]Sorbonne Université, Institut National de la Santé et de la Recherche Médicale (INSERM), Centre National pour la Recherche Scientifique (CNRS), Centre d'Immunologie et des Maladies Infectieuses, CIMI, Paris, France  [2]Unité d'Epidémiologie Moléculaire du Paludisme, Institut Pasteur du Cambodge, Phnom Penh, Cambodia  [3]Institut Pasteur, Pasteur International Network, Malaria Translational Research Pasteur International Unit, Phnom Penh, Cambodia and Paris, Paris, France  [4]ApicoLipid Team, Institute for Advanced Biosciences, Centre National pour la Recherche Scientifique (CNRS) UMR5309, Université Grenoble Alpes, Institut National de la Santé et de la Recherche Médicale (INSERM) U1209, La Tronche, France  [5]Institut des Biomolécules Max Mousseron, UMR 5247, Université de Montpellier, Montpellier, France  [6]Department of Medical Microbiology, Radboud University Nijmegen Medical Center, Nijmegen, Netherlands  [7]Commissariat à l'Energie Atomique et aux Energies Alternatives (CEA)-Université Paris Sud 11-INSERM U1184, Immunology of Viral Infections and Autoimmune Diseases (IMVA-HB), Infectious Disease Models and Innovative Therapies (IDMIT) Department, Institut de Biologie François Jacob (IBFJ), Direction de la Recherche Fondamentale (DRF), Fontenay-aux-Roses, France  [8]Institut Pasteur, Université de Paris, CNRS UMR 3528, Structural Microbiology Unit, Paris, France  [9]Université de Paris, Institut de Recherche pour le Développement (IRD), UMR 261 MERIT, Paris, France

Correspondence: romain.duval@ird.fr; cyrille.botte@univ-grenoble-alpes.fr; dominique.mazier@sorbonne-universite.fr
*†Kutub Ashraf and Shahin Tajeri contributed equally to this work

such remedies also rests with additional plant compounds that can synergize to enhance parasite killing. This claim is supported by the purported anti-malarial traditional herbal teas based on the African wormwood *Artemisia afra*, a species devoid of artemisinin (du Toit & van der Kooy, 2019). We have exploited this difference to explore this issue through comparative and extensive in vitro assessment of the inhibitory activity of infusions from both plants on *Plasmodium*.

## Results

Exposure of *P. falciparum* rings (double synchronized with 5% sorbitol at 8 h interval) to infusions from either plant for 72 h inhibited their growth in a dose-dependent manner (Fig 1A), with that of *A. annua* active at slightly lower doses. The maturation of the exposed parasites appeared to have been affected by the infusions (Fig S1). We opted to examine the biogenesis of the apicoplast and the mitochondrion as surrogates for the viability of the exposed parasites. Thus, detailed morphological examination revealed that apicoplast biogenesis was

disrupted in both *A. annua*– or *A. afra*–exposed parasites where apicoplasts failed to elongate and divide at the schizont stage as they normally do in unexposed parasites (Fig 1B). The parasite's mitochondria were also affected by both infusions (Fig 1C). These observations do not imply that the artemisinin-independent inhibitory activity is directly due to an effect on the biogenesis of these organelles. Nonetheless, preliminary data (Figs S2A–E and S3A–C) suggest that this process is affected to some extend for both organelles.

We then assessed the infusions against the pre-erythrocytic (hepatic) stages of diverse *Plasmodium* species, where part of the apicoplast metabolism (e.g., FASII activity) is indispensable (Yu et al, 2008; Vaughan et al, 2009). Exposure of primary simian hepatocytes to the infusions from the time of *Plasmodium berghei* sporozoites inoculation and for a further 48 h (spanning the time for full maturation), led to a significant dose-dependent reduction in the size and number of hepatic schizonts, with that of *A. afra* exerting its inhibitory effect at a slightly lower concentration (Fig 2A and B). Similar observations were obtained with *P. falciparum*–infected human primary hepatocytes (Fig 2C and D), a parasite with a longer hepatic development (5–6 d). Neither infusion had a cytotoxic effect

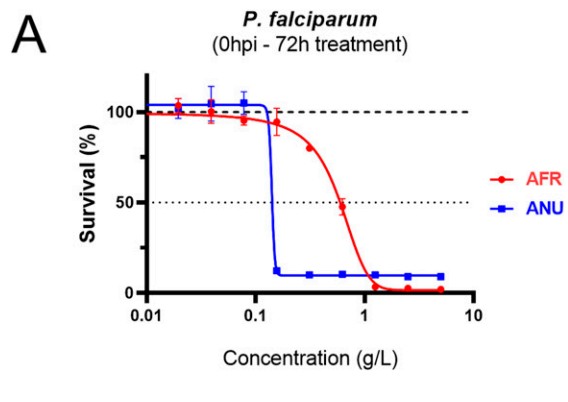

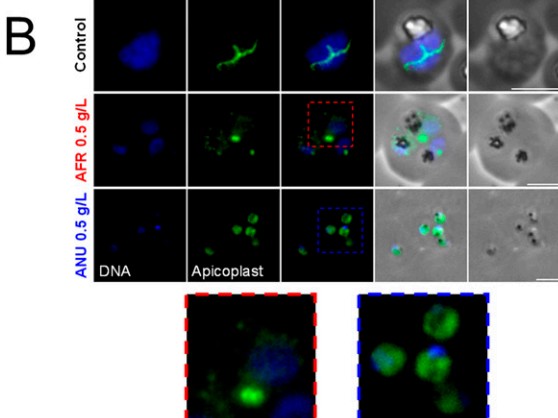

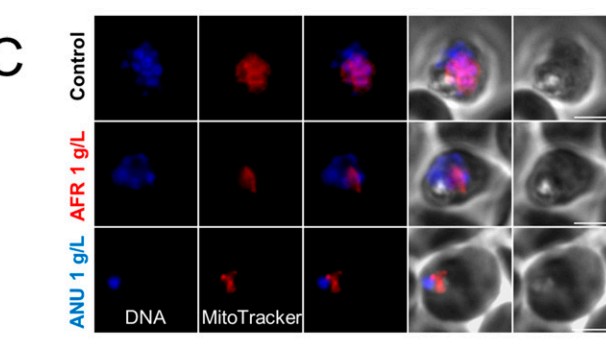

**Figure 1. Activity of *Artemisia* infusions on the *Plasmodium falciparum* asexual blood stages.**
**(A)** Survival of synchronous ring-stage parasites cultivated for 72 h in the presence of increasing concentrations of *Artemisia afra* ("AFR") or *Artemisia annua* ("ANU") infusions. Results are representative of three independent experiments. **(B)** Visualisation of apicoplasts by immunofluorescence in untreated control parasites (trophozoite stage) or those grown in the presence of the infusions (*A. afra*, trophozoite stage, red inset; *A. annua*, ring stage, blue inset). An anti-HA antibody was used to localize apicoplasts in a *P. falciparum* cell line expressing a C-terminally HA tagged version of the outer membrane triose phosphate transporter (*Pfo*TPT), a protein known to be located on the outer membrane of the apicoplast (Botté et al, 2013). Parasite nuclei were detected by Hoechst 33342 staining. Scale bar is 3 *μ*m. **(C)** Visualisation of parasite mitochondria with Mitotracker Red in infusion-exposed and control cultures. Scale bar is 3 *μ*m. Concentrations of infusions are provided as dry weight of leaves prepared in water and presented as gram per litre (g/l) (see the Materials and Methods section).

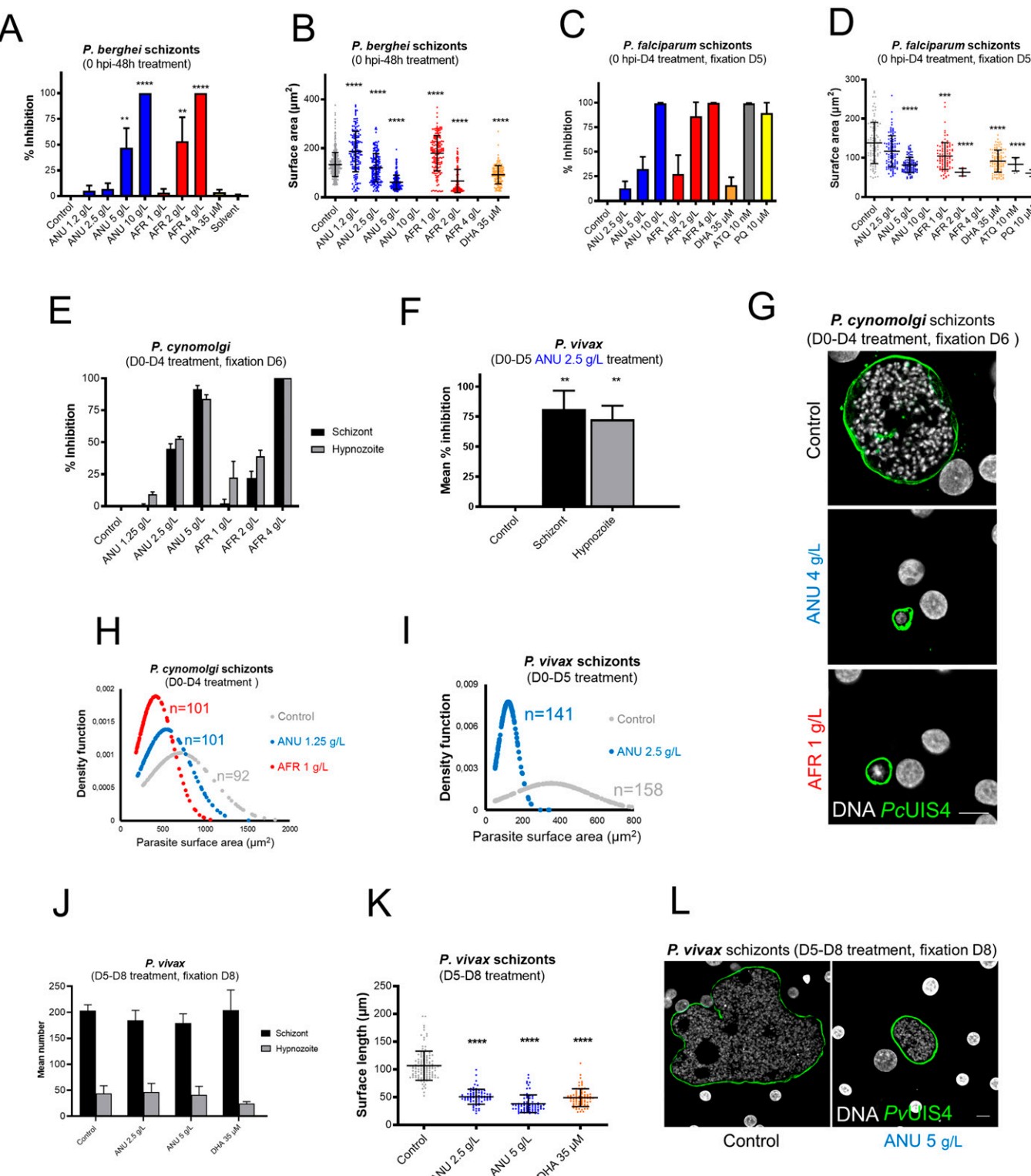

**Figure 2.   Activity of *Artemisia* infusions on the pre-erythrocytic *Plasmodium* parasites.**
**(A, B)** Quantification of intracellular *P. berghei*–GFP schizont numbers expressed as % of the mean numbers (536 per well) observed in control wells (A), and size distribution of these schizonts treated with *Artemisia* infusion and DHA (B). 10 g/l *Artemisia annua* and 4 g/l *Artemisia afra* completely cleared parasites from the cultures. **(C, D)** Impact of the exposure (D0-D4) of *Plasmodium falciparum*–infected human primary hepatocytes on parasite numbers expressed as % inhibition compared with controls (mean of 732 per well) (C), and size expressed as surface area ($\mu m^2$) (D); 10 g/l *A. annua* and 4 g/l *A. afra* cleared all *P. falciparum* schizonts from the cultures. Results are representative of three independent experiments. **(E, F)** Effect of exposure to infusions on the schizont and hypnozoite numbers in primary hepatocytes cultures of *Plasmodium cynomolgi* (E) or *Plasmodium vivax* (F) expressed as mean % inhibition. Control cultures had 120 (51 schizonts/69 hypnozoites) and 317

on the hepatocytes, nor did dihydroartemisinin (DHA, used as negative control) affect hepatic schizont numbers though their size was reduced (Fig S4A–D). We further demonstrated that exposure of sporozoites to the infusions for 1 h before their addition to the hepatocyte cultures, also led to a significant reduction in the number of hepatic schizonts for both parasite species (Fig S5A and B). Nonetheless, the viability of such exposed sporozoites, as assessed by vital staining and by their motility, was not altered (Fig S6A–D).

The assays were then extended to two parasite species, *P. vivax* and *Plasmodium cynomolgi* that produce hypnozoites. Prophylactic treatment from inoculation and for a further 4 d (D0–D4 post-inoculation) of *P. cynomolgi*–infected primary simian hepatocytes cultured in the presence of infusions significantly reduced the number of both schizonts and hypnozoites in a dose-dependent manner, by 85% for *A. annua*, and completely for *A. afra* at the highest concentrations (Fig 2E). Because of restricted number of *P. vivax* sporozoites, only the 2.5 g/l *A. annua* infusion could be tested and this had a slightly higher inhibitory effect on both schizonts and hypnozoites than that on those of *P. cynomolgi* (Fig 2F). For both *P. vivax* and *P. cynomolgi*, the remaining surviving hepatic schizonts were significantly smaller than those in control cultures (Fig 2G–I). However, exposure of *P. vivax* in human hepatocytes on days 5–8 post-infection (D5–D8 pi) to the infusions did not lead to a significant reduction in the number of schizonts or hypnozoites (Fig 2J), the schizonts showed a significant size reduction (Fig 2K and L) that was similar to that observed in parasites treated with DHA (Fig 2J and K).

The apicoplasts in the exposed sporozoites were affected in that most of the parasites displayed the dot form and some lacked an apicoplast ("void" category, Fig 3A), in contrast to those in control sporozoites where only 19% presented as a single dot with the others showing the mature elongated form (probably reflecting the presence in the salivary gland (SG) of sporozoites of different ages). For both *A. annua* and *A. afra* infusions, the shift to a high proportion of sporozoites with affected apicoplasts was dose-dependent (Fig 3A), with no concomitant change to the size, shape, or morphology of the parasite nuclei. The biogenesis of the apicoplast was also disrupted by the infusions in the hepatic stages of *P. falciparum* and *P. cynomolgi*, where an increasing fraction of parasites with "affected" apicoplasts was observed as infusion concentrations increased (Fig 3B and C). For these observations, analysis of the organellar genomes was also consistent with some inhibitory action directed against the apicoplast and the mitochondrion (Fig S2F–H). The development of apicoplasts in relapsing parasite species has been rarely recorded (Mikolajczak et al, 2015; Gural et al, 2018; Pewkliang et al, 2018), and only once for the *P. vivax* putative hypnozoite in a chimeric mouse, where it was found to consist of a single organelle elongating and then branching to form new discrete organelles that appear as a

punctate pattern. We observed a similar pattern in the hypnozoites in the *P. cynomolgi* control cultures, whereas in those exposed to the infusions (D0–D4), we noted apicoplasts with a diffuse staining pattern, implying a disrupted biogenesis (Fig 4A). Similar infusion-affected hypnozoites were also observed on D10 in *P. vivax* hepatic cultures exposed on D4–D10, although their proportion was less than that for *P. cynomolgi* (Fig 4B).

The inhibitory activity of the infusions is not primarily due to loss of the 1-deoxy-D-xylulose 5-phosphate (DOXP) pathway (demonstrated to be the only apicoplast function essential for in vitro growth of the erythrocytic parasites in lipid-rich medium) (Yeh & DeRisi, 2011; McFadden & Yeh, 2017) because similar inhibition by the infusions was observed in apicoplast-negative parasites generated via chloramphenicol (Cm) treatment that were grown in media supplemented or not with isopentenyl pyrophosphate (Yeh & DeRisi, 2011; Amiar et al, 2020) (IPP) (Fig S2A for the study design). It also appears that both infusions exhibit a strong growth inhibition at low exposition time (6 h), not rescued by the IPP addition (Fig S2B–E). The observed apicoplast loss is supported by a rapid loss overtime of the apicoplast genome in infusion-treated cultures, which parallels that of doxycycline-treated positive control cultures (Yeh & DeRisi, 2011; Uddin et al, 2018) (Fig S2F). Furthermore, quantification of mitochondrial DNA (Fig S2G) during the 48-h *P. berghei* hepatic cycle along with the microscopic imaging (Fig 2D) also revealed disruption of this organelle (Yeh & DeRisi, 2011; Uddin et al, 2018). In both control and treated conditions, parasites continued to augment their DNA and replicated during the 48-h intracellular development, but this increase in DNA was less obvious in doxycycline and infusion exposed groups (Fig 2H).

To further assess whether infusion treatments exert organelle-specific inhibition, synchronized rings were exposed to 0.5 g/l *A. annua* or *A. afra* for 24 and 48 h and apicoplast development was assessed and compared to those treated or not with 0.8 nM atovaquone (ATQ). Indeed, ATQ is a drug that is currently in use to treat human malaria, and which kills the parasite by targeting its mitochondrion, thus not directly affecting the apicoplast biogenesis. Results at 24 h show a slight but not significant effect of the ATQ treatment on apicoplast biogenesis compared to control or DMSO but both infusion treatments significantly increase the percentage of affected apicoplast with a stronger effect for *annua*-exposed parasites (Fig S3A). Apicoplast elongation and branching seemed blocked, unlike ATQ treatment, with unelongated apicoplast after *afra* treatment and apicoplast signal is diffused in the parasite cytosol with vesicles for *annua* (Fig S3A). After 48 h of treatment (Fig S3B), the percentage of parasite with affected apicoplast was even higher for the two infusions but also for ATQ, which in this case was similar to *afra* treatment. Microscopic imaging of these *P. berghei* hepatic cultures also indicated a disruption of this organelle (Fig

(205 schizonts/112 hypnozoites) hepatic parasites per well of a 96-well plate for *P. cynomolgi* and *P. vivax*, respectively. **(G)** Confocal microscopy images of exposed *P. cynomolgi* schizonts. The parasitophorous vacuole of fixed parasites were immunolabeled with anti-*Pc*UIS4 antibody. Host and parasite DNA were visualized by 4′,6-diamidino-2-phenylindole (DAPI) dye (Scale bar = 10 μm). **(H, I)** Density function plots of *P. cynomolgi* (H) and *P. vivax* (I) schizonts size distribution ("n" represents the number of schizonts for which the size was measured); the peaks show where most of the population is concentrated. **(J, K)** Quantification of *P. vivax* parasite numbers (J) and size (K) in human primary hepatocytes treated on D5-D8 pi with *A. annua* infusion. **(L)** Confocal microscopic images of treated *P. vivax* schizonts fixed at D8 pi, using an anti-*Pv*UIS4 antibody as a parasite PVM marker and DAPI for the DNA marker (scale bar = 10 μm). ANU, *A. annua* and AFR, *A. afra*. Statistical significance was determined using a one-way ANOVA followed by Krsukal–Wallis test (Dunn's multiple comparisons to control) where significance is determined by $P < 0.0001$ (****), $P < 0.0011$ (**) and $P < 0.05$ (*).

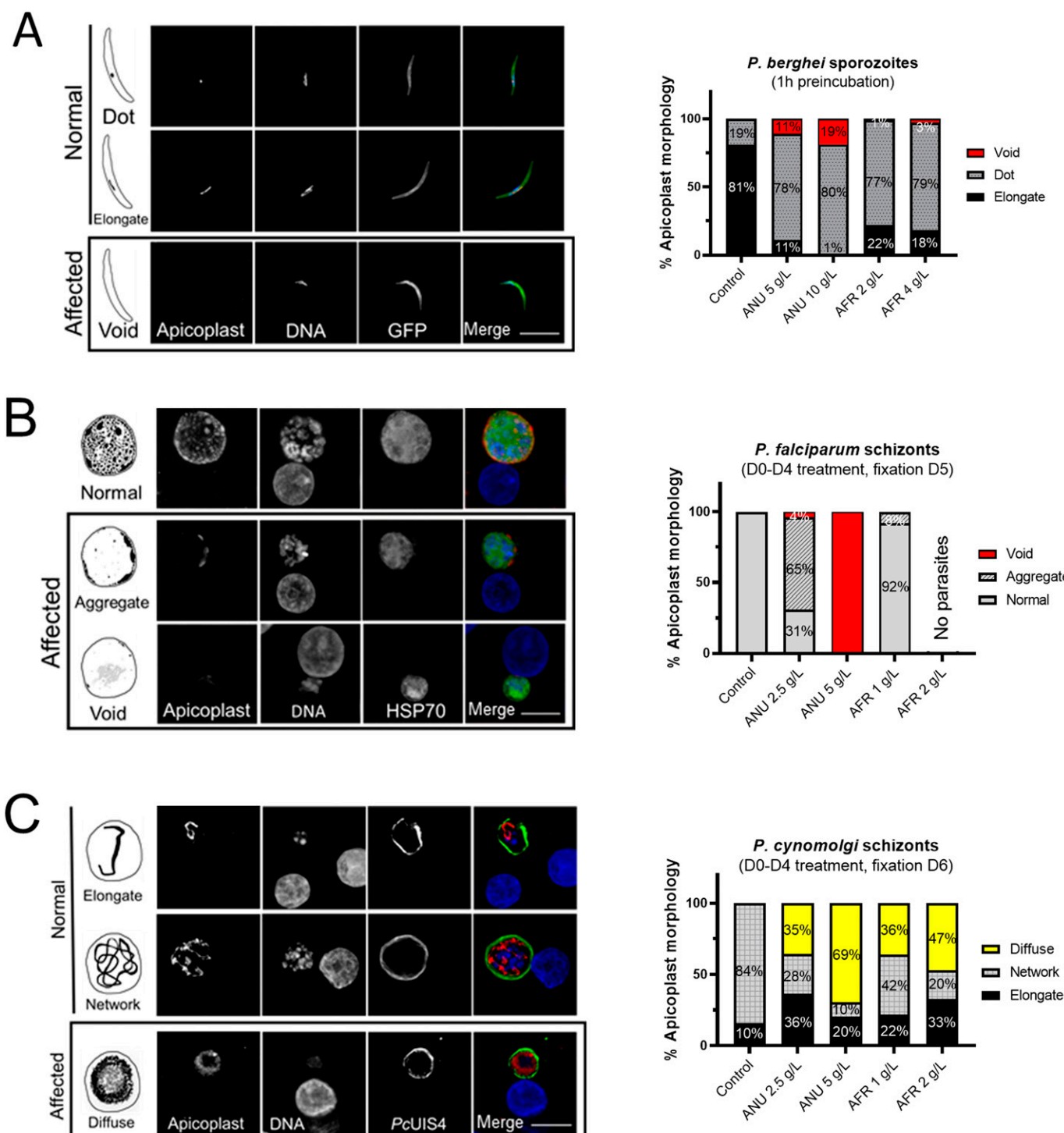

**Figure 3. Disruption of apicoplast and mitochondrial integrity in *Plasmodium* pre-erythrocytic stages by *Artemisia* infusions.**
**(A, B, C)** Schematic and confocal images of *P. berghei* sporozoites (A), *Plasmodium falciparum* schizonts (B), and *Plasmodium cynomolgi* schizonts (C) in control and treated groups. **(B)** For *P. falciparum*, the 2 g/l treatment cleared all schizonts (i.e., no parasites in panel B). The apicoplast morphologies presented in the "Affected" panels have only been observed in exposed parasites. The frequencies of the various morphologies are presented in the associated graphs. *P. falciparum* and *P. cynomolgi* parasites were detected by anti-HSP70 and *Pc*UIS4 antibodies, respectively. Apicoplasts were labelled by anti-*Py*ACP antibody (scale bar = 10 μm).

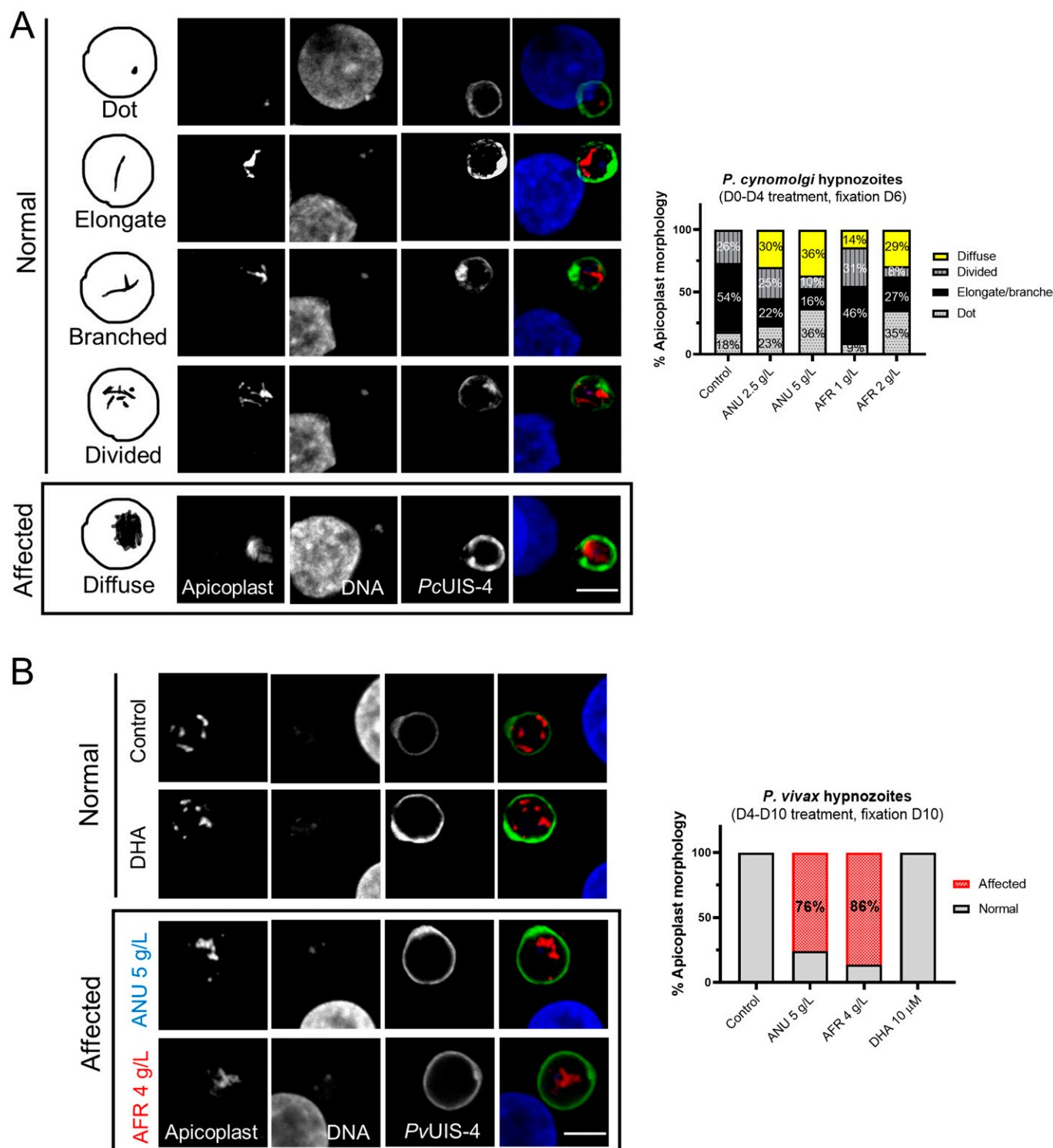

**Figure 4. Disruption of hypnozoites' apicoplast by *Artemisia* infusions.**
**(A)** Schematic (left panels) and confocal images of *Plasmodium cynomolgi* hypnozoites on D6 in cultures exposed to infusion during the early hepatic phase. In control cultures, three apicoplast morphologies could be observed, a dot-shaped apicoplast in younger hypnozoites that evolves into an elongate organelle that then branches to show a divided morphology as the hypnozoites mature. In infusion-treated cultures (D0-D4), hypnozoites with a diffuse apicoplast signal were additionally observed.
**(B)** Apicoplast morphology of *Plasmodium vivax* hypnozoites exposed to infusions from D4-D10 pi. The apicoplast morphologies presented in the "Affected" panels have only been observed in exposed parasites. **(A, B)** Quantification of the various apicoplast morphologies in 100 and 50 parasites for (A) and (B), respectively, are presented. *P. cynomolgi* and *P. vivax* were visualized by using an anti-*Pc*UIS4 and anti-*Pv*UIS4 antibodies, respectively (scale bar = 5 µm).

S3C). Together, this strongly suggests that apicoplast biogenesis might be indirectly affected by the overall death of the parasite after a longer exposure to the different treatments (*afra* or ATQ). But the stronger effect of *afra* at early stage (24 h) indicates a possible direct impact on the apicoplast. Hence, *annua* also highly affect the general development of the parasites, by blocking them at the late ring/early trophozoite stage upon 24 h. However, a high around 60% of the parasites harbour a single punctate apicoplast while ~40% harbour a disrupted/diffused apicoplast, which could also indicate an impact on the apicoplast but also and meanly a stronger impact on an apicoplast off-target that block the parasite development after short exposition to the infusion (concomitant with Fig S2D, which shown a drastic negative impact of *annua* after only 6 h of exposure).

## Discussion

The discovery of artemisinin from Chinese traditional medicine has revived interest in *Artemisia* plants as a source of treatments for diverse infectious and non-infectious pathologies beyond than malaria (de Ridder et al, 2008; Liu et al, 2009; du Toit & van der Kooy, 2019). The investigations of the in vitro anti-malarial efficacy of *Artemisia* infusions have been essentially conducted with the species *A. annua* or *A. afra*, and this exclusively on the erythrocytic stages (asexual and sexual) of *P. falciparum* parasites (Silva et al, 2012; Suberu et al, 2013; Elfawal et al, 2015). Their results are inconsistent and difficult to interpret. Data from our extensive comparative assays clearly demonstrate that infusions from either plant species can fully inhibit the multiplication and the development of both erythrocytic and pre-erythrocytic stages of various *Plasmodium sp.* in vitro. Importantly, this inhibition is independent of their artemisinin content (see Table S1 for measurement of artemisinin in both infusions), with maximal activity occurring at broadly similar infusion concentrations, without observable cytotoxic effects against the host cells. Our observations are supported by a recent study investigating the effect of these *Artemisia* infusions on asexual and gametocyte *P. falciparum* erythrocytic stages (Snider & Weathers, 2021), which shows an inhibition of the asexual growth after 48 h exposure to both infusions with a significant stronger effect for that of *A. annua*. The higher inhibition levels we observed are probably because we assessed parasite survival 72 h post-exposure (as compared to 48 h). However, we also observed a strong inhibition effect at 24 and 48 h of culture exposed to *A. annua* or *A. afra* 0.5 g/l. Of note, the inhibition of the hepatic stages appears to be mainly due to a detrimental effect on the ability of the sporozoite to achieve a productive hepatocyte infection, which also significantly led to a substantial reduction in hypnozoite numbers in the case of *P. cynomolgi* and *P. vivax*. It is likely that the detrimental effects also observed on the apicoplast in developing *P. vivax* hypnozoite will adversely affect its capacity to reactivate, although this would require further in vivo testing. It is important to note that the infusion concentrations at which we observed significant inhibition of the pre-erythrocytic stages (both sporozoites and hepatic parasites) were significantly higher (10- to 20-fold) than those needed for the *P. falciparum* erythrocytic parasites. The reason for this differential susceptibility is not known and requires further investigation.

Our in vitro observations are clearly insufficient to support claims of in vivo curative efficacy and much less for a prophylactic use for any traditional remedy from the two plants. They do however suggest the presence of hitherto unknown plant components with potent inhibitory activity. Mechanistically, PCR analysis show that some compounds present in the *Artemisia* infusions quickly and directly affect the replication of the parasite's apicoplast and mitochondria, consistent with some disruption of their biogenesis as observed by immunofluorescence imaging of these organelles in the assays conducted with several parasite species. It thus appears that one or a combination of the myriad of compounds present in the infusion might target the mitochondrion or the apicoplast directly. The direct correlation between organelle disruption or impaired development and parasite inhibition and death remains to be established.

Ultimately, the relatively similar pan-species and cross-life cycle inhibitory effects resulting from exposure to the two infusions demonstrates the presence of one or more potential lead compounds unrelated to artemisinin. This warrants systematic bio-guided screening to identify them. This is of particular interest because this might lead to a novel antimalarial class of compound that could effectively and safely prevent *P. vivax* relapses. Indeed, any drugs that could supplant 8-aminoquinolines, the only drugs currently capable of exerting hypnozoitocidal activity (Mazier et al, 2009), would greatly enhance the prospect of global malaria elimination.

## Materials and Methods

### Primary cryopreserved hepatocytes maintenance and infection

In this study, human and simian primary hepatocytes were used. Simian (*Macaca fascicularis*) hepatocytes were isolated from the liver of healthy animals using collagenase perfusion as previously described (Dembélé et al, 2011). The use of hepatocytes was approved by the Ethical Animal Committee of the Primate center, Commissariat à l'Energie Atomique et aux Energies Alternatives (permit number A 92-032-02). Cryopreserved human primary hepatocyte vials were purchased from Biopredic (Lot no. LHuf17905A) or Lonza (Lot no. HUM182641). Cells were seeded in 96-well plates (Falcon by Becton–Dickinson Labware Europe) coated with collagen I (BD Bioscience), such that a single cell layer homogenously covers each well. Cryopreserved primary human hepatocytes (Biopredic International and Lonza) were seeded generally 4 d before infection. Both human and simian hepatocytes were maintained at 37°C in 5% $CO_2$ in William's E medium (Gibco) supplemented with 10% fetal clone III serum (FCS, Hyclone), 1% penicillin–streptomycin (Gibco), $5 \times 10^{-3}$ g/l human insulin (Sigma-Aldrich), $5 \times 10^{-5}$ M hydrocortisone (Upjohn Laboratories SERB), and Matrigel (Ref. 354234; Corning). The sporozoites used to infect the cultures were purified from dissected SGs of infected mosquito (Dembélé et al, 2011). During the infection Matrigel cover was removed carefully before adding the sporozoite inoculum. The plates were then centrifuged at 900*g* for 6 min at room temperature before incubation at 37°C for 3 h. After a wash, Matrigel was added to the culture and allowed to solidify at 37°C for 45 min. The cell culture medium was changed thereafter every 24 h until cell fixation.

**List of different *Plasmodium* species used for infection of hepatocytes.**

| Parasites | Strains | Source |
|---|---|---|
| *P. berghei* | GFP ANKA | Centre d'Immunologie et des Maladies Infectieuses (CIMI), Paris, France |
| *P. falciparum* | NF54 | Department of Medical Microbiology, University Medical Centre, St Radboud, Nijmegen, The Netherlands |
| *P. cynomolgi* | M | Primate center, Commissariat à l'Energie Atomique et aux Energies Alternatives (CEA), Fontenay aux Roses, France |
| *P. vivax* | Unidentified | Symptomatic Patients, Pasteur Institute of Phnom Penh, Cambodia |

### Culturing *Plasmodium*-infected red blood cells

*P. falciparum* blood stage parasites were maintained at 2% hematocrit in 1640 RPMI-HEPES supplemented with 5% AlbuMAX II (GIBCO) and 0.25% gentamycin. Parasites were grown in sealed Perspex chambers gassed with $\beta$ mix gas (1% $O_2$, 5% $CO_2$, and 94% $N_2$) at 37°C and maintained on 48-h cycles. Cultures were tightly synchronized at ring stage using sorbitol treatment (%5 vol/vol) as previously described (Amiar et al, 2020).

### Production of *P. vivax* sporozoites

Sporozoites were obtained from mosquitoes infected by feeding on symptomatic patients presenting with *P. vivax* in Mondulkiri Province (eastern Cambodia) after obtaining their consent. The patients were managed by the medical staff based at these health facilities. Heparinized blood samples were collected by venepuncture before the initiation of treatment, and a diagnostic PCR assay was used to confirm that *P. vivax* was the only *Plasmodium* species present. The blood was centrifuged at 1,174*g* at 37°C for 5 min, and the plasma was replaced by heat-inactivated naïve human AB serum. Batches of 300–600 5–7-d-old adult female mosquitoes (*Anopheles dirus*) that were starved overnight were fed via an artificial membrane attached to a water-jacketed glass feeder maintained at 37°C. Engorged mosquitoes were maintained at 26°C and 80% relative humidity, and provided with a 10% sucrose plus 0.05% para-amino-benzoic acid solution on cotton pads. SGs of mosquitoes were dissected 14–20 d post-blood meal in L15 medium supplemented with antibiotics and Amphotericin B (Gibco).

### Production of sporozoites (other species than *P. vivax*)

*P. falciparum* was transmitted to *Anopheles stephensi* mosquitoes using an artificial membrane feeding on in vitro cultivated gametocytes, in Department of Medical Microbiology, Radboud University Medical Center, Nijmegen, The Netherlands. *P. berghei* and *P. cynomolgi* infections of *An. stephensi* mosquitoes were performed in Paris, respectively, by direct mosquito feeding on anesthetized *Pb*-GFP (Manzoni et al, 2014) infected mice, or by artificial membrane feeding on *P. cynomolgi* (strain M) infected blood from *Macaca fasicularis*. All simian procedures (infection, monitoring, and blood sampling) were performed in CEA, Fontenay aux Rose before the infected blood samples were transferred to CIMI-Paris

for mosquito infection. 2–3 wk post blood meal, *Plasmodium*-infected mosquitoes were killed with 70% ethanol. They were then washed once in Leibovitz's L-15 medium (Gibco) containing 5% fetal calf serum, 100 UI penicillin, 100 $\mu$g/ml streptomycin, and 0.5 $\mu$g/ml amphotericin B, followed by another two washes in the same medium, this time lacking serum. The mosquitos were then hand dissected under stereomicroscope and the SGs were crushed in a 1.5-ml Eppendorf tube and passed through a 40-$\mu$m filter (Cell Strainer; BD Biosciences). Sporozoites were finally counted and after dilution adjustment, directly inoculated to the cultures.

### Preparation of *Artemisia* infusion

*A. afra* (PAR, voucher ID: LG0019528 Université de Liège) and *A. annua* (LUX, voucher ID: MNHNL17733 Herbarium Luxembourg) were collected as leaves and twigs and were preserved at room temperature. Both products were protected from sunlight until infusion preparation.

The stock infusion was prepared as follows: 2.5 g *Artemisia* dried leaves and twigs were added to 25 ml of pre-boiled 25 ml commercial drinkable water (Crystalline Company) and the mixture was then boiled while being stirred for 5 min. After cooling over 10 min, the infusion was passed through a 40-$\mu$m cell strainer (Falcon, Corning Brand) to remove plant debris, and then centrifuged at 1,600*g* for 10 min to pellet any remaining fine solids, with a final filtration step over a 0.20-$\mu$m membrane filter (CA-Membrane) to obtain a fully clear solution. Filtration was done to eliminate debris produced during plant boiling. During in vitro use, debris accumulated on top of hepatocyte culture layer and made parasite quantification difficult. To our knowledge, filtration of *Artemisia* infusions is not performed by patients in endemic areas. The stock infusion (100 g/l) was stored at 4°C (short term storage) or at –20°C (long term storage). For the in vitro drug assay, the decoction was diluted to the appropriate concentration (g/l) with William's E medium (along with supplements). Frozen and freshly prepared *Artemisia* decoction samples were subjected to chemical analyses including artemisinin content quantification.

### Quantification of artemisinin in infusion: UHPLC analysis

The chromatographic apparatus consisted of Nexera X2 UHPLC system (Shimadzu) equipped with a binary pump, solvent degasser, and thermostatted column compartment. A reversed-phase

column was used for separation: Kinetex C18 (2.1 × 100 mm 1.7 $\mu m$ from Phenomenex). Mobile phase A and B consisted of water (0.1% formic acid) and acetonitrile (0.1% formic acid), respectively. The 10-min linear gradient program used was 10–100% B over 5 min, a plateau at 100% B for 2 min was used to wash the column, decreasing from 100–10% B in 0.1 min, followed by a 2.9 min post-run isocratic step at 10% B to re-equilibrate the column. The flow rate was constant at 0.5 ml/min at 25°C.

### Quantification of artemisinin in infusion: MS/MS detection

MS/MS experiments were carried out using the Shimadzu UHPLC system described above coupled to a Shimadzu LCMS-8050 triple quadrupole mass spectrometer using the multiple reaction monitoring technique operating in positive ion mode. The following parameters were used for all experiments: electrospray interface voltage was 3 kV; heat block temperature of 350°C; desolvation line temperature of 300°C; interface temperature of 300°C; drying gas flow gas was set at 5 liters/min; and nebulizing gas flow at 3 liters/min, heating gas flow rates were set at 15 liters/min. Q1 and Q3 were set to unit resolution. The dwell time was set at 1 ms for each multiple reaction monitoring transition; optimal CE values were chosen to obtain the most characteristic fragments (see below table).

### Drugs

Drugs were dissolved in different solvents according to the manufacturer's instructions which are enlisted below: atovaquone (A7986; Sigma-Aldrich), primaquine bisphosphate (160393; Sigma-Aldrich), DHA (1200520; Sigma-Aldrich), isopentenyl pyrophosphate NH4+ salt (Isoprenoids), chloramphenicol (C0378-5G; Sigma-Aldrich), and doxycycline (D9891; Sigma-Aldrich). Media with drugs were changed every 24 h during the assays.

### Immunostaining

Cultured hepatocytes were fixed using 4% PFA for 10–15 min at room temperature. The fixed samples were subjected to immune labelling. The primary antibodies (see below table) were diluted in dilution buffer (1% wt/vol Bovine Serum Albumin and 0.3% vol/vol Triton X-100 in PBS) and incubated at room temperature. Primary antibody–stained cultures were washed thrice with PBS 1X and were incubated with secondary antibody (below table) along with DAPI (1:1,000) to visualize the nuclei at room temperature for 1 h. Mitotracker (Thermo Fisher Scientific) was used to visualize parasite mitochondria.

### Quantification of Artemisinin in *A. annua* and *A. afra* infusions by UHPLC-LCMS

| Precursor (m/z) | Fragment ion (m/z) | Collision energy (eV) |
|---|---|---|
| 283.10 | 247.15 | 9 |
| 283.10 | 265.15 | 8 |
| 283.10 | 219.10 | 12 |

### Confocal microscopy and image analysis

Confocal images of immunostained cultures were taken by Leica SP8 white laser microscope controlled by the Leica Application Suite AF software at the QUANT microscopy platform (Institut du Cerveau et de la Moelle épinière, ICM) in Paris, France. Z-stack images were constructed and analyzed by Fiji package of ImageJ software. Final images were made by Microsoft Powerpoint. Schematic figures of Plasmodium apicoplast were designed by Procreate application in Apple ipad pro.

### Quantification of apicoplast morphologies

Pre-erythrocytic and hypnozoite stages: ACP-immunolabeled parasites were quantified considering the distinguishable morphology in drug-treated versus control by an individual with the aid of confocal (Leica SP8 white laser microscope) at 63X magnification. For each condition, 100 parasites were observed. Erythrocytic stage: TPT-HA–immunolabeled parasites were quantified considering the distinguishable morphology in drug-treated versus control by an individual with the aid of fluorescence microscope (Axio Imager 2_apotome; ZEISS, ×100 magnification). For each condition, between 54 and 116 parasites were observed.

### Parasite enumeration and toxicity assessment using high-content imaging

Upon fixation and immunostaining, cell culture plates were analyzed to determine the number and size of the parasites using a CellInsight High Content Screening platform equipped with the Studio HCS software (Thermo Fisher Scientific). Uninuclear hepatic parasite forms observed beyond D4 were considered to be putative hypnozoites, whereas multinucleate forms were classed as schizonts. The parasite size reduction was calculated on the average object area using the total surface area of each selected object ($\mu m^2$). The high-content imaging approach has been described previously (Bosson-Vanga et al, 2018). To assess cell toxicity of infusion for hepatic cultures, fixed plates were further scanned for the DAPI signal representing host nuclei. The analysis was based on counting of total cell nuclei.

### qPCR to quantify apicoplast and mitochondrion

To perform relative quantification of apicoplast, mitochondrial and nuclear genomes we extracted genomic DNA from *Pb*-GFP infected hepatocytes treated or not with infusion using a NuceolSpin Tissue kit (Macherey-Nagel) according to manufacturer's manual. The amount and the purity of the extracted DNA were checked by a NanoDrop ND 1000 machine (Thermo Fischer Scientific). The following gene-specific primers were designed to target genes found in each organelle or nuclear genomes:

*tufa*_ PBANKA_API00280 (apicoplast) *5′-GATATTGATTCATCTCCA-GAAGAAA-3′/5′-ATATCCATTTGTGTAGCACCAATAA-3′*,

Cytb3_PBANKA_MIT01900 (mitochondrion) *5′-AGATATATGCATG-CTACTGG-3′/5′-TCATTTGTCCCCAAGGTAAAA-3′*

**List of primary and secondary antibodies used in the study.**

| Antibody | Source | Working dilution |
|---|---|---|
| Recombinant mouse anti-*Pv*UIS4 | Noah Sather (Centre for Infectious Disease Research, Seattle, USA) | 1:20,000 |
| Mouse anti-*Pc*UIS4, polyclonal | Laurent Rénia (Singapore Immunology Network, Agency for Science, Technology, and Research, Singapore) | 1:500 |
| Recombinant rabbit monoclonal antibody against *Py*ACP | Scot E. Lindner (Huck Centre for Malaria Research, Penn State, USA) | 1:250 |
| Mouse polyclonal serum raised against *Pf*HSP70 | Centre d'Immunologie et des Maladies Infectieuses (CIMI), Paris | 1:1,500 |
| Mouse polyclonal anti-*Pf*HSP60 | Philippe Grellier (Muséum National d'Histoire Naturelle, Paris) | 1:500 |
| Rat IgG anti-HA | Roche | 1:500 |
| Alexa Fluor 488 goat anti-mouse IgG | Invitrogen | 1:500 |
| Alexa Fluor 594 goat anti-rabbit IgG | Life Technologies | 1:500 |
| Alexa Fluor 488 goat anti-Rat IgG | Life Technologies | 1:1,000 |

GFP 5′-*GATGGAAGCGTTCAACTAGCAGACC-3′*/5′-*AGCTGTTACAAACT-CAAGAAGGACC-3′*, and *CHT1*_PBANKA_0800500 (nuclear) 5′-*AACA-AAATAGAGGTGGATTT-3′*/5′-*AATTCCTACACCATCGGCTC-3′*.

The quantification of each target gene was normalized to GFP, expressed as a transgene under the *P. berghei* elongation factor 1α promoter (Janse et al, 2006). All reactions were performed on an Applied Biosystems 7300 Real-Time PCR System using the Power SYBR Green PCR Master Mix kit (Applied Biosystems), according to the manufacturer's instructions. The cycling conditions were as follows: initial denaturation at 95°C for 5 min, followed by 40 cycles at 94°C for 30 s, 56°C for 30 s, and 72°C for 30 s. The experiments were performed in triplicate. Relative quantification of target genes was calculated according to the $2^{-\Delta\Delta Ct}$ method (Schmittgen & Livak, 2008).

### Sporozoite viability assay

Freshly dissected *P. berghei*–GFP sporozoites were incubated with varying concentrations of *Artemisia* infusion for 1 h. Then the infusion was removed by centrifugation and sporozoites resuspended in culture medium. 1 μL of propidium iodide (Gibco) was added to 15,000 cells and incubated for 2–3 min. A 15-μl aliquot of sporozoite suspension was put on a KOVA Glasstic microscopic slide (KOVA International) and visualized under a microscope. Finally, dead and viable sporozoites were quantified under epifluorescence microscope.

### In vitro screening assay for sporozoite motility inhibition via video microscopy

SGs from infected mosquitoes (17–24 d post infectious blood) feeding on Swiss mice infected with *P. berghei*–GFP were isolated by hand dissection and placed into Leibovitz L15 media (11415049; Gibco) supplemented with 20,000 units penicillin–streptomycin (15140122; Gibco) and amphotericin B. Sporozoites were then released by SG mechanical disruption, filtered through a 40-μm mesh (Cell Strainer, Falcon) to remove SG debris and diluted in activation medium

(William's E medium supplemented with 10% of fetal clone III serum, 1% penicillin–streptomycin, $5 \times 10^{-3}$ g/l human insulin, and $5 \times 10^{-5}$ M hydrocortisone) to a final concentration of 400,000 sporozoites/ml and kept on ice. Aliquots of 50 μl were inoculated into wells of a 96-well plate (final amount of ~20,000 sporozoites per well). An equal volume of *Artemisia* infusion dilutions was added to the sporozoites to give a final concentration of (*A. annua* 5%, *A. annua* 10% and *A. afra* 5%) and mixed by gentle pipetting. The plate was centrifuged for 6 min at 900*g* to maximize the sporozoite settlement and immediately placed at RT for 30 min. Then, the motility of sporozoites was recorded under video microscopy Zeiss Axio Observer 7 at 40X objectives with the GFP excitation/emission filter and pictures were recorded under a frame rate of 1 frame per second with a Hamamatsu Orca Flash 4.0 V3 camera and Zen software for 3 min. Notably, the experiments were performed at 37°C. Finally, the tracking of sporozoites was further analyzed with FIJI ImageJ and the moving patterns characterized on maximum intensity z-projections. Thereby, sporozoites were classed as gliding if they moved with a circular pattern describing at least one complete circle loop during the 3 min acquisition. The percentage residual motile population was then calculated and compared with uninhibited controls (media containing an equivalent amount of water).

### SYBR Green-I–based cell proliferation assay

*Plasmodium* blood stage parasites (regular or apicoplast-free lines) are incubated in 96 well flat bottom plates, 2% hematocrit in 1640 RPMI-HEPES supplemented with 5% AlbuMAX II (GIBCO) and 0.25% gentamycin complemented with appropriate *Artemisia* infusion dilutions and with or with IPP and Cm. Parasites were grown sealed Perspex chambers gassed with β mix gas (1% $O_2$ 5% $CO_2$, 94% $N_2$) at 37°C for 72 h. New 96-well black wall flat-bottom plates are set up with 100 μl SYBR Green lysis buffer (20 mM Tris, pH 7.5; 5 mM EDTA; 0.008% [wt/vol] saponin; and 0.08% [vol/vol] Triton X-100) with freshly added SYBR Green I (1000X), and 100 μl from the cultures was transferred and mixed with the SYBR Green lysis buffer and incubated

1 h at room temperature, protected from the light. Fluorescence from each well is measured with TECAN infinite M200 plate reader (excitation: 485 nm, emission: 538 nm and integration time: 1,000 μs).

## Statistical analysis

GraphPad Prism 5 (GraphPad. Software) and Excel 2016 (Microsoft Office) were used in this study for the data analysis. All graph values are represented by means with s.d.

# Data Availability

All data are available within the manuscript.

# Supplementary Information

# Acknowledgements

K Ashraf was supported by Labex ParaFrap-IRD PhD South program (ANR-11-LABX-0024). S Tajeri acknowledges postdoctoral funding support from Fondation pour la Recherche Médicale (FRM) for the PALUKILL project. N Amanzougaghene was supported by Agence nationale de la recherche, project Plasmodrug, and Fondation Sorbonne Université. The work performed by CY Botté and C-S Arnold are supported by Fondation pour la Recherche Médicale (FRM, EQU202103012700), Agence Nationale de la Recherche, France (grant ANR-21-CE44-0010-01, Project ApicoLipidAdapt), Région Auvergne Rhône Alpes (Grant AuRA IRICE GEMELI), Finovi program (Apicolipid project), Laboratoire d'Excellence Parafrap, France (grant ANR-11-LABX-0024), LIA-IRP CNRS Program (Apicolipid project), CEFIPRA-MESRI (Project 6003-1), IDEX Université Grenoble-Alpes and MESRI PhD program fellowship. The investigations on *P. cynomolgi* were funded through a grant from the Agence Nationale de la Recherche, France (ANR-17-CE13-0025-01); IDMIT infrastructure is supported by the French government "Programme d'Investissements d'Avenir" (PIA), under grants ANR-11-INBS-0008 (INBS IDMIT). This work benefited from the support of Maison de l'Artémisia, ITMO I3M, as well as equipment and services from the CELIS cell culture core facility (Paris Brain Institute), a platform supported through the ANR grants, ANR-10-IAIHU-06 and ANR-11-INBS-0011-NeurATRIS. We would like to acknowledge the help of QUANT microscopy platform of the Paris Brain Institute specially David Akbar, Claire Lovo, and Aymeric Millécamps for their help in analysis of microscopic images and sporozoite motility videos. We thank Laurent Rénia, Noah Sather, Philippe Grellier, and Scott Lindner for the kind gift of anti-*Plasmodium* antibodies listed in the Materials and Methods section. The kind help of Sivchheng Phal and Chansophea Chhin (Institut Pasteur du Cambodge, Phnom Penh, Cambodia) with *P. vivax* experiments is sincerely thanked. We are grateful to Lucile Cornet-Vernet and Pierre Lutgen for providing us *Artemisia* plants.

## Author Contributions

K Ashraf: data curation, formal analysis, investigation, methodology, and writing—original draft, review, and editing.
S Tajeri: conceptualization, data curation, formal analysis, supervision, validation, methodology, and writing—original draft, review, and editing.
C-S Arnold: data curation, formal analysis, investigation, methodology, and writing—original draft, review, and editing.
N Amanzougaghene: data curation, formal analysis, and writing—original draft.
J-F Franetich: conceptualization, resources, supervision, investigation, visualization, methodology, and project administration.
A Vantaux: formal analysis.
V Soulard: formal analysis.
M Bordessoulles: formal analysis.
G Cazals: data curation and formal analysis.
T Bousema: formal analysis.
G-J van Gemert: formal analysis.
R Le Grand: formal analysis.
N Dereuddre-Bosquet: formal analysis.
J-C Barale: resources.
B Witkowski: supervision, funding acquisition, validation, and investigation.
G Snounou: data curation, investigation, methodology, project administration, and writing—original draft, review, and editing.
R Duval: conceptualization, data curation, formal analysis, supervision, validation, project administration, and writing—original draft, review, and editing.
CY Botté: conceptualization, data curation, formal analysis, supervision, funding acquisition, validation, investigation, visualization, methodology, project administration, and writing—original draft, review, and editing.
D Mazier: conceptualization, data curation, supervision, funding acquisition, investigation, visualization, methodology, project administration, and writing—original draft, review, and editing.

## Conflict of Interest Statement

The authors declare that they have no conflict of interest.

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
