## [Reviewer comments · Life Science Alliance]

Life Science Alliance

Artemisinin-independent inhibitory activity of *Artemisia* sp. infusions against different *Plasmodium* stages including relapse-causing hypnozoites

Kutub Ashraf, Shahin Tajeri, Christophe-Sébastien Arnold, Nadia Amanzougaghene, Jean-Francois Franetich, Amélie Vantaux, Valérie Soulard, Mallaury Bordessoulles, Guillaume Cazals, Teun Bousema, Geert-Jan van Gemert, Roger Le Grand, Nathalie Dereuddre-Bosquet, Jean-Christophe Barale, Benoit Witkowski, Georges Snounou, Romain Duval, Cyrille Botté, and Dominique Mazier

DOI: <https://doi.org/10.26508/lsa.202101237>

Corresponding author(s): Cyrille Botté, Apicolipid Team, Institute for Advanced Biosciences, CNRS UMR5309, INSERM U1209, Université Grenoble Alpes

Review Timeline:

Submission Date:	2021-09-20
Editorial Decision:	2021-09-30
Revision Received:	2021-10-23
Editorial Decision:	2021-10-25
Revision Received:	2021-11-18
Accepted:	2021-11-19

Transaction Report:

Please note that the manuscript was reviewed at *Review Commons* and these reports were taken into account in the decision-making process at *Life Science Alliance*.

Review
COMMONS

September 30, 2021

Re: Life Science Alliance manuscript #LSA-2021-01237-T

Cyrille Y Botté

Apicolipid Team, Institute for Advanced Biosciences, CNRS UMR5309,INSERM U1209, Université Grenoble Alpes

Bat. Jean Roget

Domaine de la Merci

La Tronche, Isere 38700

France

Dear Dr. Botté,

Thank you for submitting your revised manuscript entitled "Artemisinin-independent inhibitory activity of Artemisia sp. infusions against different Plasmodium stages including relapse causing hypnozoites" to Life Science Alliance. The manuscript has been seen by the original reviewers whose comments are appended below. While the reviewers continue to be overall positive about the work in terms of its suitability for Life Science Alliance, some important issues remain, in particular the conclusions drawn around apicoplast/mitochondrial targeting.

Our general policy is that papers are considered through only one revision cycle; however, given that the suggested changes are relatively minor, we are open to one additional short round of revision. Please note that I will expect to make a final decision without additional reviewer input upon resubmission.

Please submit the final revision within one month, along with a letter that includes a point by point response to the remaining reviewer comments.

To upload the revised version of your manuscript, please log in to your account: <https://lsa.msubmit.net/cgi-bin/main.plex>
You will be guided to complete the submission of your revised manuscript and to fill in all necessary information.

B. MANUSCRIPT ORGANIZATION AND FORMATTING:

Sincerely,

Eric Sawey, PhD

Executive Editor

Life Science Alliance

<http://www.lsjournal.org>

Reviewer #1 (Comments to the Authors (Required)):

Authors are addressing a significant problem in the field of malariology. There is a growing grassroots approach to self-medicating with *Artemisia annua* and *A. afra* in many communities in developing countries around the world. WHO has urged more in depth studies to validate efficacy and also explain the science behind any such efficacy. Although efforts have been made towards explaining the science behind *A. annua* and *A. afra* efficacy, much more is needed. This is a very good study focused on liver stages and including the challenging hyponozoites, so it helps towards that effort.

Upon re-review of this manuscript, I find it vastly improved and authors addressed all of my prior concerns. The supplemental data file was especially helpful. I found a few more things that would help to tidy it up:

Pg 3, ln 70-71, seems to have a typo: "...treatments, due to, are strongly discouraged variable and ..." should be restated as: "...treatments, are strongly discouraged due to variable and..."

Fig. 3 a, b, and c left panels should have some label over the fourth panel, e.g. merged.

On Fig 3b the data figure for AFR 2g/L says no parasites, yet nowhere in the text is the reason for no parasites discussed. I presume it is because as shown in Fig 2c, there was close to 100% inhibition of Pf schizonts by AFR 2 g/L. Please add a statement of explanation for Fig. 3 b "no parasites" data.

In summary, this is a very good comprehensive study that provides considerably new information on the role that both *Artemisia* species play in their therapeutic efficacy against the malaria parasite. In particular, the study shows the role of the artemisinin deficient species, *A. afra*, for its non-artemisinin antimalarial efficacy.

Reviewer #2 (Comments to the Authors (Required)):

This is the second review of the manuscript by Ashraf et al comparing antimalarial activity of infusions from *Artemisia annua* (the plant the frontline antimalarial artemisinin was derived from) and the artemisinin free *A. afra*. The finding of activity against transmission and liver stage parasites for complex mixtures derived from plant extracts is of interest to the field. Balancing that, the high concentration of a complex extract used and the unclear and still not well supported message on the apicoplast/mitochondrial targeting greatly complicates interpreting the significance of the work.

-The pitch of the manuscript has largely changed from saying the extracts kill the parasite by targeting the apicoplast and mitochondria, to, the function and replication of these organelles is damaged by the drug. To me, the bulk of the evidence now supports that parasite death is likely impacting on the apicoplast/mitochondrion. If there are direct apicoplast/mitochondrial targeting components, there is virtually no way that it can be confirmed with these extracts or using the data from this paper. Still, the messaging suggests there may be components that directly target these organelles:

'It thus appears that one or a combination of the myriad of compounds present in the infusion (Supplementary Data Information) might target the mitochondrion or the apicoplast directly. The direct correlation between organelle disruption or impaired development and parasite inhibition and death remains to be established.'

This is impossible to be sure about this from this data and detracts from the important message of activity at multiple life stages.

-Referring to Suppl Figure 2

'For these observations, analysis of the organellar genomes were also consistent with some inhibitory action directed against the apicoplast and the mitochondrion (Supplementary Data Fig, 2f-h).'

'Mechanistically, PCR analysis show that some compounds present in the *Artemisia* infusions quickly and directly affect the replication of the parasite's apicoplast and mitochondria, consistent with some disruption of their biogenesis as observed by immunofluorescence imaging of these organelles in the assays conducted with several parasite species.'

Doxycycline is used as a control drug in this experiment, presumably because it has apicoplast targeting delayed death activity at lower concentrations. The concentration used in this experiment is high enough to kill 50% of blood stages with only 48 hours treatment (Uddin et al, Dahl et al), meaning that this is likely through a non delayed death mechanism of action. The data presented in Suppl Fig 2 f, g and h suggests that Doxycycline is also killing liver stages over a short time-period at a similar rate to the extracts. This raises the question of whether the drug is working through an apicoplast targeting delayed death mechanism of action, or whether the concentration is causing non-specific and faster killing. Given reduced mitochondrial and nuclear DNA, it looks like Doxycycline might be killing the parasite through a non-apicoplast targeting mechanism. With all three drugs showing a reduction in nuclear DNA at 48 hours, these data mainly support that the primary target of the extracts is not the apicoplast or mitochondria.

-Reviewer 1 raises the point about high extract concentrations used in this study, and highlights a 20-fold higher initial extraction concentration made (100g/L) compared to Ref 7 (5g/L). The response to reviewers indicates that at least the asexual stage assays were done at a similar range to Ref 7 (0.02g/L to 5g/L, Fig1a). However, it is important to note that Ref7 used their extract at a 36-fold lower concentration than the starting material (0.137g/L). While this lies in the range of the blood stage IC50 curve, the extracts in this paper under review were not active against blood stage growth until >0.1g/L. The majority of stage specific experiments used >10 fold higher final extract concentration before inhibition was seen compared to Ref7. I think reviewer 1's point is valid and relatively high concentrations of extract are used in this study. This was not discussed or addressed in a meaningful way in the rebuttal or the text.

Reviewer #1 (Comments to the Authors (Required)):

Authors are addressing a significant problem in the field of malariology. There is a growing grassroots approach to self-medicating with Artemisia annua and A. afra in many communities in developing countries around the world. WHO has urged more in depth studies to validate efficacy and also explain the science behind any such efficacy. Although efforts have been made towards explaining the science behind A. annua and A. afra efficacy, much more is needed. This is a very good study focused on liver stages and including the challenging hyponozoites, so it helps towards that effort. Upon re-review of this manuscript, I find it vastly improved and authors addressed all of my prior concerns. The supplemental data file was especially helpful. I found a few more things that would help to tidy it up:

Pg 3, ln 70-71, seems to have a typo: "...treatments, due to, are strongly discouraged variable and ..." should be restated as: ...treatments, are strongly discouraged due to variable and...

Corrected.

Fig. 3 a, b, and c left panels should have some label over the fourth panel, e.g. merged.

This has been added.

On Fig 3b the data figure for AFR 2g/L says no parasites, yet no where in the text is the reason for no parasites discussed. I presume it is because as shown in Fig 2c, there was close to 100% inhibition of Pf schizonts by AFR 2 g/L. Please add a statement of explanation for Fig. 3 b "no parasites" data.

A phrase has been added to the Figure 3 legend (highlighted in yellow).

In summary, this is a very good comprehensive study that provides considerably new information on the role that both Artemisia species play in their therapeutic efficacy against the malaria parasite. In particular, the study shows the role of the artemisinin deficient species, A. afra, for its non-artemisinin antimalarial efficacy.

Reviewer #2 (Comments to the Authors (Required)):

This is the second review of the manuscript by Ashraf et al comparing antimalarial activity of infusions from Artemisia annua (the plant the frontline antimalarial artemisinin was derived from) and the artemisinin free A. afra. The finding of activity against transmission and liver stage parasites for complex mixtures derived from plant extracts is of interest to the field. Balancing that, the high concentration of a complex extract used and the unclear and still not well supported message on the apicoplast/mitochondrial targeting greatly complicates interpreting the significance of the work.

-The pitch of the manuscript has largely changed from saying the extracts kill the parasite by targeting the apicoplast and mitochondria, to, the function and replication of these organelles is damaged by the drug. To me, the bulk of the evidence now supports that parasite death is likely impacting on the apicoplast/mitochondrion. If there are direct apicoplast/mitochondrial targeting components, there is virtually no way that it can be confirmed with these extracts or using the data from this paper. Still, the messaging suggests there may be components that directly target these organelles: 'It thus appears that one or a combination of the myriad of compounds present in the infusion (Supplementary Data Information) might target the mitochondrion or the apicoplast directly. The direct correlation between organelle disruption or impaired development and parasite inhibition and death remains to be established.'

This is impossible to be sure about this from this data and detracts from the important message of activity at multiple life stages.

We do not agree that raising the possibility that one or both organelles might be a target for some of the compounds present in the infusion detracts from the message, which is an effect by some compounds other than artemisinin. The data provides some justification for this speculation, and we were careful to point out that this was a mere possibility and that this will need to be confirmed or disproved in future studies.

-Referring to Suppl Figure 2 'For these observations, analysis of the organellar genomes were also consistent with some inhibitory action directed against the apicoplast and the

mitochondrion (Supplementary Data Fig, 2f-h).' Mechanistically, PCR analysis show that some compounds present in the Artemisia infusions quickly and directly affect the replication of the parasite's apicoplast and mitochondria, consistent with some disruption of their biogenesis as observed by immunofluorescence imaging of these organelles in the assays conducted with several parasite species.'

Doxycycline is used as a control drug in this experiment, presumably because it has apicoplast targeting delayed death activity at lower concentrations. The concentration used in this experiment is high enough to kill 50% of blood stages with only 48 hours treatment (Uddin et al, Dahl et al), meaning that this is likely through a non delayed death mechanism of action. The data presented in Suppl Fig 2 f, g and h suggests that Doxycycline is also killing liver stages over a short time-period at a similar rate to the extracts. This raises the question of whether the drug is working through an apicoplast targeting delayed death mechanism of action, or whether the concentration is causing non-specific and faster killing. Given reduced mitochondrial and nuclear DNA, it looks like Doxycycline might be killing the parasite through a non-apicoplast targeting mechanism. With all three drugs showing a reduction in nuclear DNA at 48 hours, these data mainly support that the primary target of the extracts is not the apicoplast or mitochondria.

We agree that at the doxycycline concentrations used parasite killing has taken place over the 48 h (the actual IC₅₀ for this drug on the *P. berghei* ANKA-GFP hepatic stages is not necessarily similar to that for the *P. falciparum* erythrocytic stages where IC₅₀ values can vary substantially between different lines). Nonetheless, Fig S2H that the parasite genomic DNA has increased during the 48h cycle (each sampling point is normalized to control) when the cultures were exposed to the three drugs, and not reduced as the reviewer stated (see above underlined), while apicoplast and mitochondrial DNAs decreased. Thus, although *Artemisia* infusions prevented parasite growth, qPCR quantification data show that nuclear DNA replication occurs in the surviving parasites but both apicoplast and mitochondrial genome replication is impaired, which is an indication that specific targeting of these organelles might be operating.

-Reviewer 1 raises the point about high extract concentrations used in this study, and highlights a 20-fold higher initial extraction concentration made (100g/L) compared to Ref 7 (5g/L). The response to reviewers indicates that at least the asexual stage assays were done at a similar range to Ref 7 (0.02g/L to 5g/L, Fig1a). However, it is important to note that Ref7 used their extract at a 36-fold lower concentration than the starting material (0.137g/L). While this lies in the range of the blood stage IC₅₀ curve, the extracts in this paper under review were not active against blood stage growth until >0.1g/L. The majority of stage specific experiments used >10 fold higher final extract concentration before inhibition was seen compared to Ref7. I think reviewer 1s point is valid and relatively high concentrations of extract are used in this study. This was not discussed or addressed in a meaningful way in the rebuttal or the text.

We are perplexed by this comment concerning the concentration of the infusions (in g/L) to which the parasites were exposed. Indeed, our stock solution was more concentrated than that in Snider et al. (100 g/L versus 5 g/L), but this is immaterial since we diluted this stock to yield the working solutions (0.01 to 10 g/L). We observed near total inhibition of the *P. falciparum* cultures at concentrations between 0.1 and 0.2 g/L (see Fig. 1A) which corresponds rather well with the significant inhibition observed by Snider at 0.137 g/L (expressed as a percentage change in parasitaemia after 48hr by these authors). We do agree that the concentrations needed for the inhibition of the pre-erythrocytic stages are higher than those for the erythrocytic parasites (at least for *P. falciparum* the only parasite that we could culture in blood). We have added a note to highlight this in the Discussion (at the end of the first paragraph). However, providing an explanation for this difference awaits future investigations.

October 25, 2021

RE: Life Science Alliance Manuscript #LSA-2021-01237-TR

Dr. Cyrille Y Botté
Apicolipid Team, Institute for Advanced Biosciences, CNRS UMR5309,INSERM U1209, Université Grenoble Alpes
Bat. Jean Roget
Domaine de la Merci
La Tronche, Isere 38700
France

Dear Dr. Botté,

Thank you for submitting your revised manuscript entitled "Artemisinin-independent inhibitory activity of Artemisia against Plasmodium including hypnozoites". We would be happy to publish your paper in Life Science Alliance pending final revisions necessary to meet our formatting guidelines.

AWMR:

- please upload your main and supplementary figures as single files
- please note that titles in the system and manuscript file must match
- please make sure the author order in your manuscript and our system match
- please be sure that the author contribution matches the system and manuscript file
- please add callouts for Figures S2A-E; S3A-C; S4A-D; S5A, B; S6A-D to your main manuscript text
- in the legend for Figure 2, panel L appears to be indicated with the letter I
- It appears that the simian primary hepatocytes were isolated in the lab. If so, please included a statement detailing ethical approval.
- Please indicate in the legends what the various asterisks mean for Figures 2, S1 and S3. Similar to what was done for Figure S5.
- The information included in the separate "Supplemental Material" file should be incorporated into the main text. This seems be a collection of Methods and additional discussion points, along with Table S1 and additional References.

A. FINAL FILES:

B. MANUSCRIPT ORGANIZATION AND FORMATTING:

Sincerely,

November 19, 2021

RE: Life Science Alliance Manuscript #LSA-2021-01237-TRR

Dr. Cyrille Botté

Apicolipid Team, Institute for Advanced Biosciences, CNRS UMR5309,INSERM U1209, Université Grenoble Alpes

Dear Dr. Botté,

Thank you for submitting your Research Article entitled "Artemisinin-independent inhibitory activity of Artemisia against Plasmodium including hypnozoites". It is a pleasure to let you know that your manuscript is now accepted for publication in Life Science Alliance. Congratulations on this interesting work.

DISTRIBUTION OF MATERIALS:

Again, congratulations on a very nice paper. I hope you found the review process to be constructive and are pleased with how the manuscript was handled editorially. We look forward to future exciting submissions from your lab.

Sincerely,

Eric Sawey, PhD

Executive Editor

Life Science Alliance

<http://www.lsajournal.org>